# Characterization of *Moringa oleifera* Leaf Powder Extract Encapsulated in Maltodextrin and/or Gum Arabic Coatings

**DOI:** 10.3390/foods10123044

**Published:** 2021-12-08

**Authors:** Toyosi T. George, Ayodeji B. Oyenihi, Fanie Rautenbach, Anthony O. Obilana

**Affiliations:** 1Department of Food Science and Technology, Faculty of Applied Sciences, Cape Peninsula University of Technology, P.O. Box 1906, Bellville 7535, South Africa; 219385408@mycput.ac.za; 2Functional Foods Research Unit, Faculty of Applied Sciences, Cape Peninsula University of Technology, P.O. Box 1906, Bellville 7535, South Africa; oyenihia@cput.ac.za; 3Oxidative Stress Research Centre, Faculty of Health and Wellness, Cape Peninsula University of Technology, P.O. Box 1906, Bellville 7535, South Africa; RautenbachF@cput.ac.za

**Keywords:** food fortification, gum Arabic, maltodextrin, microencapsulation, *Moringa*

## Abstract

The encapsulation of bioactive-rich plant extracts is an effective method of preventing their damage or loss of activity during processing and storage. Here, the techno-functional properties of microcapsules developed from *Moringa oleifera* leaf powder (MoLP) extract (core) with maltodextrin (MD), gum Arabic (GA), and a combination (MDGA) (coatings) were assessed. The bulk and tap density were 0.177, 0.325 and 0.297 g/mL and 0.13, 0.295 and 0.259 g/mL for GA, MD and MDGA microcapsules, respectively. Flowability properties of microcapsules indicated an intermediate flow except for GA which had a poor flow. The moisture content of the microcapsules ranged from 1.47% to 1.77% with no significant differences (*p* > 0.05) observed. All the microcapsules had high water solubility (86.35% for GA to 98.74% for MD and 90.51% for MDGA). Thermogravimetric analyses revealed that encapsulation enhanced the thermal stability of the core material. The X-ray diffraction analysis revealed that the microcapsules and extracts have an amorphous nature, which was validated by the surface morphology analysis that showed amorphous, irregular, and flake-like attributes except for MDGA microcapsules which had slightly spherical and agglomerated surfaces. The Fourier Transform Infra-Red spectra of the microcapsules showed the presence of C-O and O-H aromatic rings as well as amine groups. New spectra were observed at 1177, 1382 and 1411 cm^−1^ for MDGA, MD and GA, respectively, after encapsulation, which connotes a slight modification in the chemical structural pattern after encapsulation. Storage stability tests (28 days at 4, 25 and 40 °C) showed that the microcapsules were most stable at 4 °C and the stability differs significantly (*p* ≤ 0.05) with coating material type and temperature with MDGA showing better storage stability than others. Altogether, the attributes of the MDGA microcapsules were comparatively better than either MD or GA alone. The present data, therefore, demonstrate an effective encapsulation process for MoLP extract that can serve as fortificants in processed food products where MoLP may be used.

## 1. Introduction

*Moringa oleifera Lam* is a well-known plant that originated from the Himalayan and middle Eastern regions of the world [1,2] but has now become naturalised to many countries of the world [3]. The plant is reportedly rich in important bioactive and phytochemical compounds that have the potential for use in the development of functional foods [4,5]. The presence of these bioactive compounds has heightened the conventional use of the plant as a viable inclusion in foods for fortification and enrichment purposes, improving the overall functional importance of foods [5,6]. The bioactive compounds in *Moringa oleifera* leaf powder (MoLP) were established to be potent against free-radical molecules that cause oxidative stress, resulting from an imbalance between free radicals and antioxidants in the body [3,7,8]. Examples of such major bioactive compounds in MoLP extracts are flavonoids and phenolic acids [9,10]. However, these compounds are highly susceptible to damage during processing under high temperatures [11] and storage conditions [12]. In addition, some of these bioactive compounds have astringent and bitter tastes which may make foods unpalatable [13].

The stability and storage life of the bioactive compounds in MoLP have been of concern to researchers over the years because of their susceptibility to degradation, which many have attributed to the presence of unsaturated bonds present in their phytochemicals [14]. This feature makes the bioactive compounds prone to damage at high temperatures and exposure to light [2,4]. A procedure which has been proposed to mitigate the impact of these and other factors is encapsulation [15]. Encapsulation is a procedure for protecting core or active compounds in protective matrices known as carriers or coating materials, to produce microcapsules with desirable properties [15,16]. This procedure has been employed previously for the protection of flavours, colorants, essential oils, bioactive compounds as well as other important active compounds [16,17,18,19,20]. It has also been reported to be important for the retention and stabilization of active compounds in foods; shielding sensitive and beneficial phytochemical compounds from degradation that might occur during processing and from environmental conditions during storage; prevention of reaction of active compounds with food products as well as controlling their rate of release [20,21,22,23,24]. Moreover, encapsulation is believed to improve the bioavailability and dissolution properties of bioactive compounds in foods [25].

Different coating materials have been used over the years for the encapsulation of plant extracts. The most common of these coating materials is maltodextrin, a colourless modified starch with high water solubility and low viscosity at high concentrations as well as good film-forming properties [25,26,27]. To further improve the stability and the encapsulation efficiency of microcapsules produced with maltodextrin coatings, its combination with other coating materials for example gum Arabic, was recommended [28,29]. Gum Arabic, is a long-chain polysaccharide gum with high solubility, good emulsifying and stabilizing attributes and low viscosity in aqueous solution [26,30]. Maltodextrin in combination with gum Arabic has been used in the encapsulation of spent coffee ground extract [31], chlorophylls [32], *M. oleifera* oil [33], eggplant extract [28], and anthocyanin from *Ipomea batatas* [34] to mention a few and has shown great potential in all of these studies. Therefore, we hypothesize that these combined coatings will improve the stability of MOLP extract.

For the application and inclusion of microcapsules of *M. oleifera* leaf powder extracts in food matrices, the understanding of their physical, functional, morphological, thermal, structural and storage behaviour is essential. To the best of our knowledge, this is the first study on the encapsulation of MoLP extracts in maltodextrin and gum Arabic matrix using freeze-drying. Therefore, the objective of this study is to develop microcapsules of MoLP and evaluate their techno-functional (physical, functional, thermal and storage properties.

## 2. Materials and Methods

### 2.1. Plant Raw Material and Chemicals

*Moringa oleifera* leaf powder (MoLP) was obtained from SupaNutri Graff-Reinet, South Africa, and kept at room temperature prior to extraction. Maltodextrin coating of dextrose equivalent (DE) 16.5–19.5 was obtained from Sigma Aldrich, Germany, through Sigma Aldrich, South Africa. Gum Arabic, acasia ash <4% was obtained from Thermofisher, Kandel, Germany through Industrial Analytical, Cape Town South Africa. Ethanol for extraction was obtained from United Scientific, Goodwood, Cape Town South Africa. All reagents used were of analytical grade and all equipment used for testing was operated under Good Lab Practices (GLP) conditions.

### 2.2. Extraction of Bioactive Compounds from Moringa Oleifera Leaf Powder (MoLP)

The extraction of the bioactive (core) compounds from MoLP was carried out using the method of Dadi et al. [11] with slight modification. The MoLP (10% *w*/*v*) was dissolved in 60% ethanol and the mixture was agitated for 24 h in a laboratory shaking incubator (Incotec shake model 355, Labotec, Cape Town, South Africa) at 120 rpm and a temperature of 27 °C. The mixture was then centrifuged using an Eppendorf centrifuge 5810 R (Hamburg, Germany) at 10,000 rpm for 10 min at 4 °C. The supernatant was decanted and residue discarded. The supernatant was concentrated by removing ethanol using a rotary evaporator (BÜCHI Labortechnik R-200 Rotavapor, Flawil, Switzerland) set at a temperature of 40 °C. The extract obtained was freeze-dried using a Virtis genesis 25EL freeze-drier (Gardiner, New York, NY, USA) for 48 h to obtain dried extract powder, which was stored until further analysis.

### 2.3. Preparation of Coating Material for Encapsulation

The method described by Cilek et al. [26] was adopted with some modifications. Maltodextrin (MD) and gum Arabic (GA) were selected as coating materials separately for MD and GA microcapsules or combined (7.5:2.5) for MDGA microcapsules. These materials were dispersed in respective centrifuge tubes containing distilled water (10% *w*/*v*) and allowed to mix overnight (18 h) using a standing shaker, model 205 (Labotec, Cape Town, South Africa) for complete hydration. The coating material solutions were mixed using a high-speed homogeniser (IKA—WERIE, Stanfen, Germany) at 9500 rpm for 15 min and allowed to stand for at least 1 h before they were used in the preparation of the MoLP extract microcapsules.

### 2.4. Preparation of MoLP Extract Microcapsules

The method described by Cilek et al. [26] was employed with some modification. Freeze-dried MoLP extract (1 g) was re-dissolved in distilled water by vigorous mixing, this was then mixed with the coating material solution (10% total solids in the solution) using a high-speed homogeniser for 10 min at 9500 rpm after which it stood for 1 h. The mixtures were subjected to ultrasonication in an ice bath for 10–20 min at 100 W power and 20 kHz frequency with a 50% pulse rate using a titanium probe (3.8 mm diameter) using the ultrasound sonicator (BANDELIN electronic, Berlin Germany). After ultrasonication, the mixture stood for 5 h at room temperature for self-alignment before it was transferred into a freeze-dryer (Hanil, Gimpo, South Korea) for 48 h to obtain the dried MD, GA and MDGA *Moringa oleifera* leaf powder (MoLP) extracts microcapsules.

### 2.5. Moisture Content

The moisture contents of the microcapsules were determined by weighing 1 g of the sample in a hot air oven set at 105 ± 1 °C for 3 h (AOAC, 2000). This was performed in triplicate for precision. The weight of the dried microcapsules was determined as expressed as the percentage of initial weight minus final weight divided by the initial weight in the equation below.
(1)$$\mathrm{Moisture}\;\mathrm{content}=\frac{W1-W2}{W1}\times 100$$
where *W*1 is the weight of the initial sample and *W*2 is the weight of the final sample obtained after the drying.

### 2.6. Hygroscopicity

Hygroscopicity analysis was carried out to determine the amount of moisture the *Moringa oleifera* microcapsule was able to absorb from the storage environment. The Hygroscopicity test was performed using the method described by Sun et al. [35] with slight modification. A total of 1 g of *Moringa oleifera* microcapsule samples was placed in Petri dishes before being placed in containers that were well-sealed and contained a saturated solution of Sodium chloride (NaCl) at a working temperature of 25 °C and relative humidity of 75%. After fourteen days, the samples were weighed and the hygroscopicity was determined as the mass of moisture absorbed per 100 g of the dry *M. oleifera* microcapsules.
(2)$$\mathrm{Hygroscopicity}=\frac{\left(W2-W1\right)\;}{W1}\times 100$$
where *W*1 is the weight of the initial sample and *W*2 is the weight of the final sample obtained after the hygroscopicity treatment.

### 2.7. Bulk and Tapped Density

Bulk and tapped density measurements were carried out using the method reported by Dadi et al. [36]. For the bulk density determination, 5 g of the MoLP microcapsule was placed in a 25 mL measuring cylinder, the volume occupied by the microcapsules was then recorded while for tapped density, the measuring cylinder with 5 g MoLP microcapsules was tapped steadily on a laboratory tabletop, until a consistent volume was obtained. The bulk density is expressed mathematically as the ratio of the weight of the microcapsule to the volume occupied in the graduated cylinder (g/mL).
(3)$$\mathrm{Bulk}\;\mathrm{Density}\;(g/\mathrm{mL})=\frac{\mathrm{weight}\;\mathrm{of}\;\mathrm{microcapsule}}{\mathrm{volume}\;\mathrm{occupied}}$$

### 2.8. Flowability

The flow properties of the microcapsules were evaluated using the method described by Mahdavi et al. [37] with slight modifications. These were carried out using Hausner’s ratio, Carr’s index and the angle of repose. The angle of repose was measured by using a fixed laboratory funnel and a digital angle ruler (DXN-360, DigiX New, Tokyo, Japan). The Hausner’s ratio and Carr’s index were determined using the initial data obtained from the bulk and tapped density values in the equation below.
(4)$$\mathrm{Hausner}’\;s\;\mathrm{ratio}=\frac{\mathrm{Tapped}\;\mathrm{density}}{\mathrm{Bulk}\;\mathrm{density}}$$
Carr’ s index = (Tapped density-Bulk density)/(Tapped density) × 100(5)
Angle of repose (°) = tan^−1^ (H/R) (6)
where H is the height of the pile and R is the radius at the base.

### 2.9. Water Absorption and Solubility Index

The water absorption index (WAI) of the *M. oleifera* microcapsules was determined using the method reported by Dadi et al. [10] with slight modifications. Sample (1 g) was placed in a 50 mL centrifuge tube of known weight. Distilled water (25 mL) was then added, after which the mixture was mixed vigorously using a mechanical shaker at 35 °C for 30 min. The resulting mixture was then centrifuged at 4000 rpm for 4 min. The supernatant was decanted and kept apart while the hydrated gel was weighed to determine the WAI using the equation below.
(7)$$\mathrm{Water}\;\mathrm{absorption}\;\mathrm{index}=\frac{\mathrm{weight}\;\mathrm{of}\;\mathrm{hydrated}\;\mathrm{gel}}{\mathrm{weight}\;\mathrm{of}\;\mathrm{an}\;\mathrm{initial}\;\mathrm{sample}}$$

The water-solubility Index (WSI) of the *M. oleifera* microcapsules was determined by pouring the supernatant from the WAI determination, in an aluminium pan of known weight. The supernatant was then dried overnight in an oven set at 105 °C. The WSI was expressed as the ratio of dried supernatant to the ratio of the initial sample multiplied by 100%.
(8)$$\mathrm{Water}\;\mathrm{solubility}\;\mathrm{index}=\frac{\left(\mathrm{weight}\;\mathrm{of}\;\mathrm{dried}\;\mathrm{supertant}\right)}{\mathrm{weight}\;\mathrm{of}\;\mathrm{initial}\;\mathrm{sample}\;}\times 100$$

### 2.10. Surface Morphology by the Scanning Electron Microscopy

The morphological attributes of MoLP extract microcapsules were examined using a Zeiss MERLIN scanning electron microscope (Carl Zeiss Microscopy, Oberkochen, Germany). The method described by Lei et al. [38] with slight modification was used. Briefly, 5 mg microcapsules were mounted on stubs of carbon double-sided adhesive tape that was mounted on aluminium SEM stubs. It was then coated with gold to a thickness of 10 nm. Samples were evaluated by SEM at a voltage of 15 kV and SEM images were recorded at different magnifications.

### 2.11. Fourier Transform Infrared (FT-IR) Spectroscopy

Fourier transform infrared spectroscopy was used in the identification of chemical groups and bonding present in the microcapsules, the core and coating materials. The FT-IR spectra for all the samples were obtained using a Jasco FT-IR 4000/6000 series spectrometer, Jasco Oklahoma, USA. A 5 mg sample was mixed with solid potassium bromide (KBr) powder after which the transmittance was recorded at wavelengths of 400–4000 cm^−1^.

### 2.12. Crystallinity Pattern by X-ray Diffraction

The crystalline and amorphous structure of the microcapsules, core and coating materials were determined using the XRD—D8 advanced diffractometer (Bruker AXS Berlin Germany). The radiation was generated at a tube voltage and current of 40 kV and 40 mA respectively, with a 2Ɵ angle of 5 to 80° and a measurement time of 0.5 s/step.

### 2.13. Thermal Behaviour

The thermogravimetric analysis (TGA) of the microcapsules was performed using a Shimadzu thermogravimetric analyser (Model TGA 50, Shimadzu Corporation, Kyoto, Japan) using the method of Ballesteros et al. [31]. Briefly approximately 5 mg sample was placed in a small aluminium plate and heated from 30 °C to 800 °C at a heating rate of 10 °C/min and under a nitrogen atmosphere at a flow rate of 50 cm^3^/min. Weight loss was measured using the data generated from the analysis.

### 2.14. Storage Stability Test

To evaluate the storage stability test of the microcapsules and the *M. oleifera* extracts, all samples were stored at 4, 25 and 45 °C for 4 weeks. The Folin–Ciocalteu method as reported by Dadi et al. [10] was adopted and used to determine the loss in the total phenolic content (TPC) weekly in six experimental replicates. The percentage TPC loss was calculated after 4 weeks.

### 2.15. Statistical Analysis

Experiments were conducted in at least triplicates and values were reported in mean ± standard deviation. Statistical analysis was conducted using a one-way Analysis of variance (ANOVA) (IBM SPSS version 26, 2020). Mean values were compared using Duncan’s least significant difference (LSD) and considered statistically significant when *p* < 0.05.

## 3. Results and Discussion

### 3.1. Moisture Content

The moisture content of the microcapsules ranged from 1.47–1.77% (Table 1). The GA microcapsules had the highest moisture content (1.77%) compared to the MDGA microcapsules with the lowest moisture content (1.47%). This is similar to the observation of Nawi et al. [34] where MDGA, MD and GA were used in the encapsulation of anthocyanins from *Ipomea batatas*. The values reported for moisture for all samples indicated that different coating materials used did not significantly (*p* > 0.05) impact the moisture content of the microcapsules. Similar observations were also made by Kuck and Noreña [39] and Dadi et al. [10] in a study on the encapsulation of grape phenolic extract using gum Arabic and polydextrose (a modified starch similar to maltodextrin) and encapsulation of *M. stenopetala* extracts, respectively. All moisture contents reported in the current study are lower than the values reported by these authors for freeze-dried microcapsules and within the ranges reported by Premi and Sharma [33] for microcapsules of *M. oleifera* seed oil coated with maltodextrin and gum Arabic. The values indicated that all developed microcapsules can be stored for prolonged periods without degradation related to high moisture content.

### 3.2. Hygroscopicity

The hygroscopicity of the microcapsules ranged from 11.13–15.86% (Table 1). The hygroscopicity of MD microcapsules was significantly (*p* ≤ 0.05) higher than MDGA, an indication that coating had a significant effect on how much moisture microcapsules absorbed from their surrounding during storage. This observation follows similar trends reported by Dadi et al. [10]. The MD microcapsules have previously been reported to possess high moisture absorbing capacities which may be ascribed to the presence of the OH functional group [40,41]. The MDGA microcapsules were the most desirable as they possessed the lowest hygroscopicity which may contribute to the stability of microcapsules over a long storage period. The lower hygroscopicity may have resulted from chemical interaction between the structure of the combined coating material and the MoLP extract. The hygroscopicity for GA did not differ significantly (*p* > 0.05) from the MD microcapsules. Since MDGA had the lowest hygroscopicity, it is preferred over MD and GA microcapsules as low hygroscopicity improves storage stability and decreases their susceptibility to moisture-related degradation.

### 3.3. Water Solubility and Absorption Index (WAI and WSI)

The solubility properties of microcapsules are related to their reconstitution attribute [42]. The values for the water solubility index for the microcapsules produced in this study, ranged from 86.35–98.74% (Table 1), indicating that all microcapsules exhibited high solubility properties in water. There were significant differences (*p* ≤ 0.05) among all the samples, which implies that the coating material type and combination, influence the solubility properties of the microcapsules. The MD microcapsules had the highest WSI (98.74%) compared with the GA microcapsules with the lowest (86.35%). The high solubility index from MD samples can be attributed to the high solubility property of maltodextrin [43]. The MDGA microcapsules had a value in between the two samples coated with MD and GA, the difference may be ascribed to the blending of the two different coating materials which may have resulted in intermolecular and chemical interaction. A similar observation was made when the maltodextrin-gum Arabic mixture was used as a coating in the encapsulation of eggplant extract, by Sarabandi et al. [28]. In addition to this, the MDGA microcapsules had a lower WSI value than the value reported for MD microcapsules, which was similar to trends reported in two recent studies using similar carriers [10,28]. The reported values for the samples indicated that encapsulation had the potential to increase the solubility properties of plant extracts in water. Since MD microcapsules had the highest solubility, it is preferred in terms of solubility in water over the MDGA and GA microcapsules.

Similarly, the water absorption index (WAI) was determined to measure the ability of the microcapsules to take up water. The results of WAI, range from 0.15–0.23 g (Table 1). The MDGA microcapsules had the least WAI while the GA microcapsules had the highest WAI. There was no significant difference (*p* > 0.05) between MDGA and MD microcapsules. This may be ascribed to the high ratio of maltodextrin to gum Arabic in the former, this result showed that the interaction between the two coatings did not noticeably change the ability of the two samples to retain water. Although the GA microcapsules were observed to have a much higher WAI, this is likely due to the high molecular weight and the hydrophilic nature of gum Arabic which is typical of many natural gums that are long-chain polysaccharides [44].

### 3.4. Bulk and Tapped Density

The bulk density of the various microcapsule samples ranged from 0.177 to 0.325 g/mL (Table 1). There were significant differences (*p* ≤ 0.05) in the bulk density among all microcapsules indicating that coating material composition affected the bulk density. The MDGA samples had a bulk density value of 0.297 g/mL compared to the MD and GA with bulk densities of 0.325 g/mL and 0.177 g/mL, respectively. The relatively high bulk densities translate to ease of storage with less space requirement. Previous reports showed that high bulk densities indicate low amounts of air in the microcapsules, thereby decreasing the chances of air-related degradation such as oxidation [45,46]. The bulk densities observed are consistent with findings by Dadi et al. [10]. In the current study, microcapsules produced from GA had the lowest bulk density which may be ascribed to the viscous property of GA, this indicates that a larger storage container is required for GA-coated samples. Similar observations were reported when maltodextrin, maltodextrin-gum Arabic, and gum Arabic were used as coating materials in the encapsulation of eggplant extracts by Sarabandi et al. [28]. The differences were ascribed to the relatively low viscosity of gum Arabic, which may have resulted in the formation of large microcapsule particles. The differences may also be attributed to the inherent chemical composition of the coating, their molecular weight, and the internal structural bond exhibited by the coating materials [37]. The irregular and amorphous structure of the microcapsules resulting from the freeze-drying process may have contributed to significant differences observed among all three samples with different coating materials. The values obtained show that coating material affects the bulk density of all developed microcapsules and MD microcapsules seem to possess better bulk density than others.

Tapped density values ranged from 0.126 to 0.295 g/mL (Table 1), with significant differences (*p* < 0.05) between all three samples indicating that coating material composition affected the tapped density. As stated earlier, the structural pattern, molecular weight of coating material, and their inherent viscous properties contribute to the tapped density of microcapsules produced. The results reported for the three samples follow the same trend reported by Sarabandi et al. [28] where the highest tapped density was reported in the sample coated with MD alone. The combination of GA and MD resulted in a slight decrease in tapped density indicating the influence of coating material combination on this characteristic of MDGA coated microcapsules which were significant (*p* ≤ 0.05). The sample encapsulated with GA had the lowest tapped density value which may be ascribed to its viscous property. Hence, the tapped density values reported indicated that coating material has a significant effect on the density of produced microcapsules which may, in turn, affect the space required for packaging and oxidative stability due to the presence of airspaces.

### 3.5. Flowability

Flowability is an important parameter used in predicting and deciding the condition of processing, formulation, packaging, and transportation of products in the food industry [29]. To evaluate the flow properties of the produced microcapsules, the Hausner’s ratio, Carr’s index, and angle of repose were measured, this is because a single parameter cannot be used in predicting the flow properties of granular particles such as microcapsules [37].

The Hausner’s ratio values ranged from 1.10 to 1.41, Carr index values from 9.23 to 28.83%, and the angle of repose from 32.01 to 34.98° (Table 1). The values reported in the current study are within the ranges reported by Premi and Sharma, [33] It is believed that if Hausner’s ratio, Carr’s index, and the angle of repose values are greater than 1.25, 25%, and 45, respectively, the resultant product can be described as poor flowing material [37,46] therefore, the results in the present study indicate that the MD and MDGA microcapsules have medium flow properties. This may be attributed to the amorphous structural pattern of the microcapsules as well as the low moisture content which may have affected the cohesive and frictional forces among the microcapsules [10]. There were no significant differences (*p* ≤ 0.05) for Hausner’s ratio and Carr’s index value of both MDGA and MD microcapsules. The Hausner’s ratio and Carr’s index values (1.41 and 28.83 respectively) reported for the GA microcapsules established that GA microcapsules have poor flow property, although the angle of repose value seems lower than the benchmark for such classification. Hence, in terms of flowability, MDGA and MD microcapsules showed great promise as they possess flow properties suitable for use in many food applications, which may be ascribed to the low viscous attribute of the coating materials used as well as the lower surface areas of microcapsules produced.

### 3.6. Surface Morphology of MoLP Microcapsules

The surface structures of the microcapsules developed from different coating materials as elucidated using a Scanning Electron Microscope (SEM) are shown in Figure 1 with all the samples exhibiting different external morphologies. The MDGA microcapsules have a somewhat smooth exterior with few dented surfaces and conspicuous agglomeration that may have resulted from the freeze-drying procedure. This is similar to the smooth surface morphology of maltodextrin-hydroxyl pectin blend microcapsules reported earlier by [10]. The MDGA microcapsules were larger than others, the large particle size may be due to low process temperature and combination of different coating materials [42]. In comparison with MDGA, the MD microcapsule show an amorphous, flake and glass-like structure typical of many microcapsules developed from maltodextrin [47,48]. This appearance is slightly different from the MDGA microcapsules, as the appearance of MDGA may likely be due to the interaction that has occurred during the preparation of constituent coatings (maltodextrin and gum Arabic). A similar surface morphology to MD microcapsules was observed in freeze-dried MD microcapsules by Chranioti et al. [18]. Generally, most microcapsules developed using freeze-drying are characterised by an amorphous glass-like appearance with dented [26] and shrivelled surfaces which have been ascribed to the effect of low-temperature drying.

The appearance of the GA microcapsules was an irregular-shaped, brittle structure, amorphous microcapsules with some dented surface and flake-like appearance, that may have resulted from freeze-drying [31,42]. Similar in appearance to these microcapsules reported is the characteristic morphology of microcapsules of beetroot extracts coated with gum Arabic by Chranioti et al. [18]. The appearance of all microcapsules only differed slightly, although the MDGA microcapsules seemed to have a surface and particle morphology perceived to be better than others, as they appeared to have an external morphology indicative of efficient coating of the core compound which is further validated by the high encapsulation efficiency. Although a structurally amorphous and irregular-shaped external morphology of microcapsules may affect their storage stability through exposure to degradation caused by oxidation [30], while their release and solubility properties are believed to be enhanced by their amorphous structural outlook [11]. The amorphous surface morphology of freeze-dried microcapsules may impact their flow behaviour, causing resistance to flow which may have resulted from the large surface areas of the microcapsules observed in the SEM images.

### 3.7. Structural Properties of MoLP Microcapsules by FTIR

Fourier transform infrared (FTIR) spectroscopy was used to elucidate the presence of some characteristic functional groups in the microcapsules as well as to study the potential interaction between the coating materials and the core compound.

The characteristic absorption band for MoLP extract (Figure 2) spectrum appeared at 3226 cm^−1^ (O-H stretching vibration, also ascribed to the presence of bioactive polyphenols), 2924 cm^−1^ (C–H stretching band and CH2 vibration of aliphatic hydrocarbons), 1667 cm^−1^ (C=C aromatic stretching ring, characteristic of phenolic compounds) [48,49,50,51,52,53]. The two peaks at 1599 and 1511 cm^−1^ were identified, as amide II stretches, C=O and C=C aromatic stretching [54]. Those at 1352 and 1053 cm^−1^, were identified as amines, C-O and C-O-H bending respectively [32]. The peak observed at 519 cm^−1^ may likely be due to the presence of aliphatic halogens (iodides) [55]. Generally, the peaks in the extracts further confirmed the presence of organic and phenolic acid-like compounds.

The spectrum of pure maltodextrin (Figure 2) presented absorption bands at 3288 cm^−1^ (O-H stretching), 1369 cm^−1^ (CH_2_ asymmetrical bending) and 1016 cm^−1^ (C–O and C-O-H stretching and bending) which agrees with some previous literatures [32,54,55,56]. The spectrum of gum Arabic presented absorption bands at 3294 cm^−1^ (O–H broadband), 1625 cm^−1^ (C=O stretching and N-H bending) and 1018 cm^−1^ (C–C and C-O stretching band) [32].

For all microcapsules (Figure 2), the presence of O-H stretching bands was observed at 3292, 3294 and 3300 cm^−1^, although the intensity of these peaks decreased after encapsulation.

This observation is consistent with the previous findings of Kang et al. [32], peaks present in the core and coating materials were present while new peaks were formed, further confirming that encapsulation occurred. The broad peaks at 2960 and 2949 cm^−1^ for MDGA and GA microcapsules respectively can be ascribed to the presence of COOH and C-H bands, which are characteristic peaks of MoLP bioactive compounds that may have been entrapped in the microcapsules [54]. New peaks were identified in MDGA, MD and GA microcapsules at 1177 cm^−1^ (C–O aromatic ringed amines group), 1382 cm^−1^ (aromatic C=O, C=N, NH and C=C aromatic stretching vibrations) and 1411 cm^−1^, respectively [53,54,57,58]. The formation of these new peaks may likely be due to stretching caused by high-frequency mixing of the ultrasound cavitation.

In all, the new peaks obtained from all microcapsules indicated that the encapsulation of the active compound was successful for all selected coating materials but with minimal structural modifications that may have resulted from high energy mixing caused by ultrasound cavitation.

### 3.8. The Crystallinity of Microcapsules Using X-ray Diffraction

X-ray diffraction analysis measures the structural pattern, characteristics and crystallinity of constituent compounds present in the microcapsules [56]. The measured crystallinity may influence the stability of the formed microcapsules [33]. The XRD patterns of MDGA, MD and GA microcapsules, and pure maltodextrin and gum Arabic coatings, as well as the MoLP extracts, are presented in Figure 3. The XRD patterns showed a very low degree of crystallinity and diffused broad peaks for all samples. This indicates that all samples were amorphous and disordered with noise around, a characteristic attribute of microcapsules coated with maltodextrin and gum Arabic [31,32].

The MoLP extract, MDGA, MD, GA microcapsules, maltodextrin and gum Arabic coatings all showed a defined but low crystalline peak 2θ around 21° and a broad diffused peak around 39°. MoLP extract presents two amorphous peaks around 21° and 38°, the low degree of crystallinity of these peaks indicates its amorphous nature.

For the microcapsules, MDGA, MD and GA presented broad diffused peaks that are conspicuous around 19° 21° and 22°, respectively, which may be indicative of total entrapment of the core in the coating material. The dense, noisy pattern of the diffractogram shows that the microcapsules were amorphous which may have resulted from the absence of the attribute needed for defined crystallinity during the process of freeze-drying [43]. All peaks reported around 20° indicated the presence of the core in the coating materials. The peaks obtained for MD and GA microcapsules were similar to the diffused peaks reported earlier for mango extract microcapsule coated with MD and GA respectively by Cano-Chauca et al. [43]. Broad diffused peaks were also observed around 34° for MDGA, 35° for MD and 40° for GA microcapsules. Both MD and GA coatings had two broad peaks around 19° and 20°, respectively (Figure 3). This observation is consistent with broad peaks obtained for both coating materials in the study of Ballesteros et al. [31].

The peaks as seen in the pure coating material mirrored those found in the microcapsules which indicated that the core was completely entrapped in the coating materials. Other broad, diffused peaks were observed around 35° and 42°, respectively for pure maltodextrin and gum Arabic coatings.

In all, the XRD pattern displays that the extract and the microcapsules are structurally amorphous which was consistent with the SEM analysis results. An amorphous crystalline property may aid the release of the active compound and enhance the water solubility [10] however, this may affect the storage stability of the microcapsules [28].

### 3.9. Thermal Stability of MoLP Microcapsules

The thermogravimetric analysis (TGA) of the dried MoLP extracts, the coating materials (MD and GA) and the microcapsules, were performed to elucidate the thermal stability of the products (Figure 4a–d). Generally, in thermal behaviour analysis, the temperature at which the highest weight loss is observed is described as the maximum degradation temperature [58]. As seen in Figure 4a and when compared with Figure 4b–d, the major structural changes and weight loss caused by temperature are related to the coating material used. Pure gum Arabic and maltodextrin presented a two-step weight loss, the first weight loss of 7.3 and 7%, respectively, were observed around 63 and 70 °C, respectively, similar to the observations of da Cruz et al. [59] for pure gum Arabic. The second weight loss of 47 and 54% occurred around 290 and 282 °C respectively. Similar observations were made for pure gum Arabic by da Cruz et al. [59] and maltodextrin by Ballesteros et al. [31].

The TGA plot of MDGA, GA and MD microcapsules (Figure 4b–d) indicates a three-step weight loss of compounds present in all samples. The first weight loss of 5, 3 and 7.9% in MDGA, MD and GA respectively occurred around 115, 108 and 97 °C, these losses can be attributed to moisture or water loss from the microcapsules at these high temperatures [31]. The presence of OH bond was confirmed by the FTIR analysis at around 3200 cm^−1^, which may be attributed to the surface moisture present in the microcapsule. The second weight loss of about 59% (MDGA), 56% (MD) and 57% (GA) occurred around 341 °C. The decomposition here may be ascribed to the degradation of phenolic compounds that were entrapped in the coating material. The third weight loss was observed around 743, 764 and 738 °C with losses of 18.83, 18.52 and 14.76% for MDGA, MD and GA, respectively, which may have signalled the destruction of all the core material portions with complex antioxidant attributes. For the un-encapsulated *M. oleifera* leaf powder extracts, there was a three-step weight loss. The first loss of 6.5% occurred around 70 °C, which may likely be due to gradual evaporation of surface moisture present in the freeze-dried extract as well as the degradation of heat-labile bioactive compounds that deteriorate below boiling temperature (100 °C).

There was a significant weight loss of about 45% between 135 and 320 °C which may have resulted from the thermal decomposition of polyphenols and other bioactive compounds present in the extracts with different structural patterns. A similar observation was made for the second stage of weight loss for eucalyptus extracts by Gullón et al. [49]. The observations above show that the coating of extracts had a significant protective effect on the core. The observations also indicate that encapsulation of *M. oleifera* bioactives enhances their thermal stability, this is in agreement with Ballesteros et al. [31] who observed that encapsulated extract of ground spent coffee was more thermally stable than spent coffee ground extracts. The last stage weight loss of the MDGA, MD and GA samples occurred around 753, 764 and 738 °C, respectively. In general, the degradation or weight loss that occurred in all encapsulated samples were similar and shows higher thermal stability than un-encapsulated extracts. In addition to this, all microcapsules showed similar stability at high temperatures, with the MDGA and MD microcapsules being more thermally stable.

### 3.10. Storage Stability Test

The storage stability based on the total phenolic content (TPC) of the microcapsules produced with different coating materials was studied over 28 days at different temperatures (4, 25 and 40 °C). The MoLP extract was used as a control. At 40 °C, there was a rapid decrease in the TPC of all the samples in the first 21 days (Figure 5). This is in tandem with the observation of Dadi et al. [10], where the rate of decrease in TPC observed for all samples was faster in the first 20 days during 90-day storage. After 28 days, there was approximately a 50% loss in the TPC present in MoLP extracts at 40 °C, whereas there was a total loss of 26.88%, 40.14% and 42.45% observed in the encapsulated samples (MDGA, MD and GA, respectively) (Figure 5A–D). The significant difference (*p* < 0.05) observed between the encapsulated samples and the extracts can be ascribed to the protective effect of the coating material. Since the primary role and goal of encapsulation is to shield the core compounds from degradation caused by environmental factors so that they retain their bioactivity. The storage at this temperature (40 °C) shows that encapsulation resulted in the fulfilment of this crucial goal. It is also important to mention that the use of combined coating materials (MDGA) resulted in decreased degradation (26.88%) during 28 days of storage when compared with the other samples. The amounts of total phenols lost in all microcapsules were significantly lower than those reported for *M. oleifera* extracts coated with tragacanth gum stored at 35 °C for 35 days reported by Castro-López et al. [60]. The increased retention of total phenols observed in MDGA over this period may have resulted from the lower amount of surface polyphenols present in these microcapsules as well as the high encapsulation efficiency of this combined coating material. The lower loss of TPC observed in MDGA is similar to the observations of Sansone et al. [61] and Dadi et al. [10], where high retention of TPC was observed after storage. There was no significant difference (*p* > 0.05) observed in the quantity of TPC lost after 28 days between MD and GA microcapsules. The rate of loss observed indicated that with time, there is a reduction in the amount of the TPC present in stored products.

At 25 °C, a loss of 27.05% was observed after 28 days in the TPC of the MoLP extract while only a loss of 11.11, 16.41 and 22.53% of TPC in MDGA, MD and GA, respectively. This further shows the significance of encapsulation as crucial procedure in conserving beneficial compounds in foods and beverages during storage. The MDGA microcapsules recorded the lowest loss at this temperature.

At 4 °C, minimal losses in TPC were recorded after 28 days. A 4.40, 7.84 and 8.46% loss were reported for MDGA, MD and GA, respectively, indicating that at low temperature, minimal losses of phenolic content occur. Only 10% of the initial TPC present in the extract powder was lost at 4 °C implying that this temperature or lower temperatures enhance the storage stability of dried extracts. In all, the stability of the microcapsules during storage was affected by temperature and the coating material used. Furthermore, a progressive decline in the TPC was observed with respect to time (days) for all samples which were similar to a previous observation by Osamede Airouyuwa and Kaewmanee, [62]. This study shows that low-temperature storage at 4 °C will minimize loss and that the combined use of coating material (MDGA) offers increased stability and as such is recommended.

## 4. Conclusions

The *M. oleifera* leaf powder extract was successfully encapsulated in maltodextrin, gum Arabic and a blend of maltodextrin–gum Arabic. The effectiveness of the encapsulation process was further confirmed by the FTIR, TGA and storage stability analyses which revealed superior characteristics with the microcapsules compared with the extract alone. The structural analyses of microcapsules by the XRD and SEM indicated they are amorphous with irregular structural patterns. In addition, all microcapsules exhibited better stability than the extract at high temperatures when assessed by thermogravimetric analysis but the MDGA microcapsules were the most stable during the storage period. The MD microcapsules, however, showed better bulk density attributes relative to others while those of MDGA and MD had better flow behavior. Overall, the MDGA microcapsules may be the most preferred due to the comparatively higher stability and better functional properties than both MD and GA microcapsules. These developed microcapsules can form an inclusion in traditional foods because they maintain the intactness of the bioactive and nutritional components in MoLP extract during processing and storage, therefore, improving the benefits of such foods.

## Figures and Tables

**Figure 1 foods-10-03044-f001:**
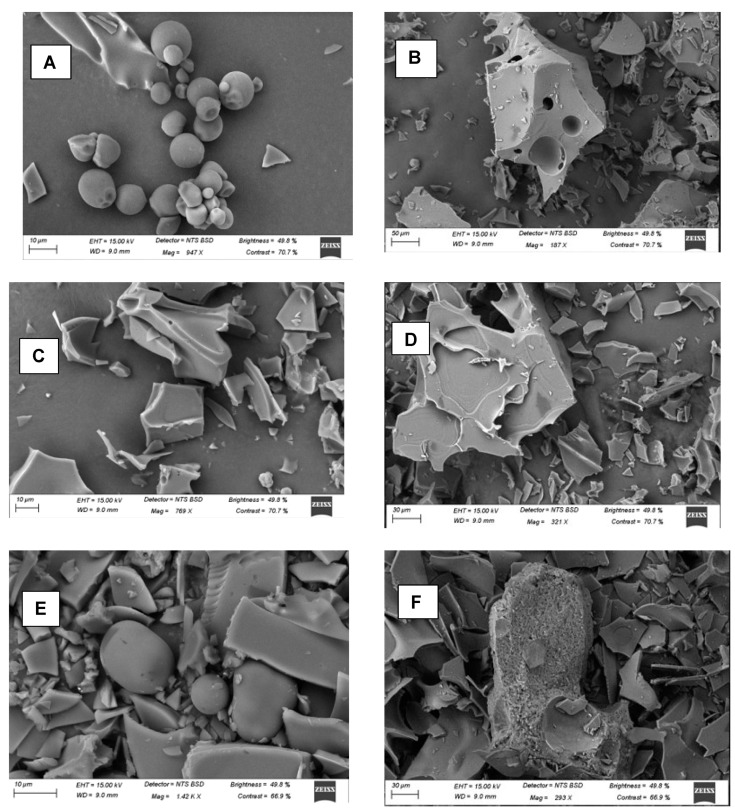
Morphological structures of microcapsules (**A**,**B**) MDGA (**C**,**D**) MD (**E**,**F**) GA.

**Figure 2 foods-10-03044-f002:**
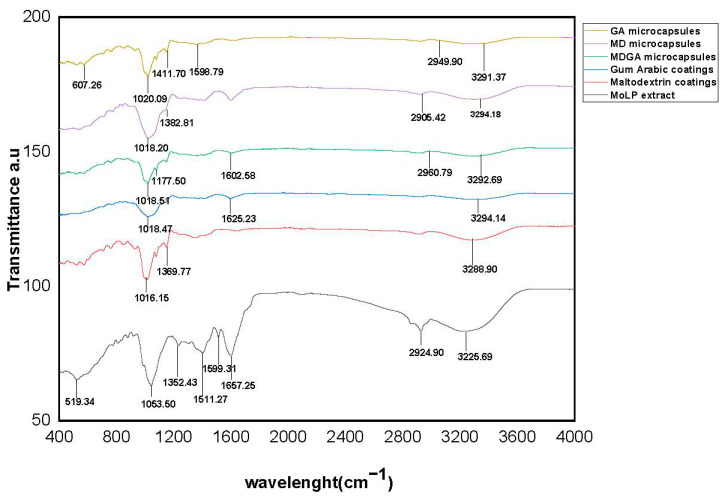
FTIR spectra of maltodextrin, gum arabic, *M. oleifera* leaf powder extracts, microcapsules of *M. oleifera* leaf powder with maltodextrin, gum arabic and maltodextrin-gum Arabic mixture.

**Figure 3 foods-10-03044-f003:**
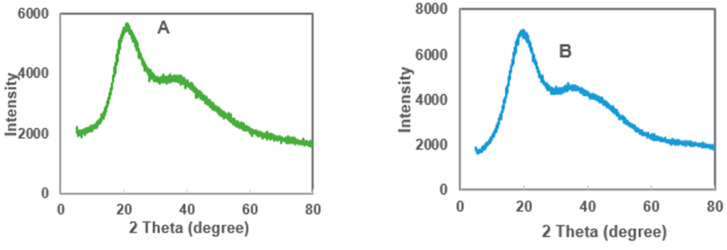
X-ray diffractogram of (**A**) *M. oleifera* extract (**B**) *M. oleifera* microcapsules in MDGA (**C**) *M. oleifera* microcapsule in MD (**D**) *M. oleifera* microcapsule in GA (**E**) MD (**F**) GA coatings.

**Figure 4 foods-10-03044-f004:**
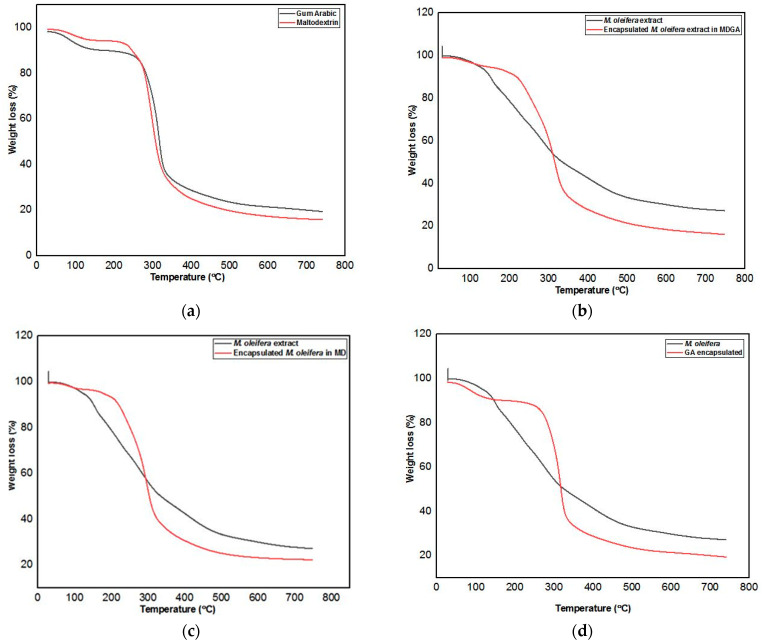
TGA (**a**) plot of pure coating materials (**b**) MDGA encapsulated and *M. oleifera* extract (**c**) MD encapsulated, and *M. oleifera* extract (**d**) GA encapsulated and *M. oleifera* extract.

**Figure 5 foods-10-03044-f005:**
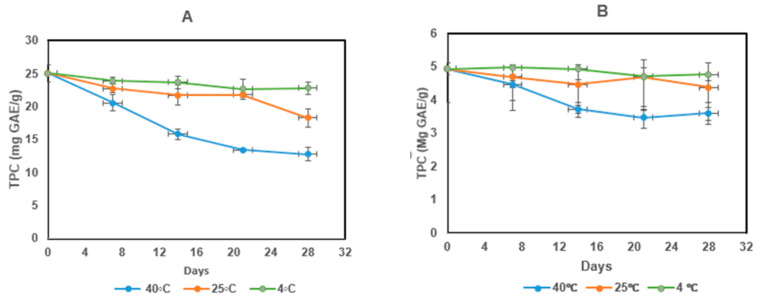
Total polyphenol contents at different temperatures for (**A**) *M. oleifera* extract (**B**) MDGA microcapsules (**C**) MD microcapsules (**D**) GA microcapsules.

**Table 1 foods-10-03044-t001:** Physical properties of MoLP extract microcapsules.

Parameters	Samples		
	MDGA	MD	GA
Moisture content (%)	1.53 ± 0.18 ^a^	1.47 ± 0.24 ^a^	1.77 ± 0.35 ^a^
Hygroscopicity (%)	11.13 ± 0.92 ^a^	15.86 ± 2.22 ^b^	14.35 ± 1.59 ^b^
Water Solubility Index (%)	90.51 ± 2.19 ^a^	98.74 ± 1.05 ^b^	86.35 ± 4.78 ^c^
Water Absorption Capacity (g)	0.15 ± 0.04 ^a^	0.17 ± 0.02 ^a^	0.23 ± 0.03 ^b^
Bulk density (g/mL)	0.30 ± 0.01 ^a^	0.33 ± 0.00 ^b^	0.18 ± 0.00 ^c^
Tapped density (g/mL)	0.26 ± 0.00 ^a^	0.30 ± 0.00 ^b^	0.13 ± 0.00 ^c^
Hausnier ratio	1.15 ± 0.03 ^a^	1.10 ± 0.06 ^a^	1.41 ± 0.06 ^b^
Carr’s index (%)	12.55 ± 2.25 ^a^	9.23 ± 0.26 ^a^	28.83 ± 2.95 ^b^
Angle of Repose (◦)	34.54 ± 0.76 ^a^	32.01 ± 0.32 ^b^	34.98 ± 0.69 ^a^

Data are presented as mean ± standard deviation, different superscripts in the same row indicate significant differences (*p* < 0.05) using Duncan’s test. MDGA: Maltodextrin and gum Arabic coated mixture sample; MD: maltodextrin coated sample; GA: Gum Arabic coated sample.

## Data Availability

All data from this study have been reported in the manuscript.

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
