# Peer review of "Characterization of Moringa oleifera Leaf Powder Extract Encapsulated in Maltodextrin and/or Gum Arabic Coatings"

_foods, 2021, doi:10.3390/foods10123044_

Round 1

Reviewer 1 Report

Characterization of Moringa oleifera leaf powder extract encapsulated in maltodextrin and/or gum Arabic coatings

This is a good manuscript with interesting results. Authors performed a classical approach to preparation and characterization of capsules. Discussion of results and conclusions can be improved by providing a few analyses based on the cumulative evidence derived from the various techniques used. Other points are as follows:

Abstract

How the amorphous. Irregular and flake-like structures were assessed?

Materials and methods

2.4 Firstly, a dispersed system was prepared by creating a homogenized material that cannot be regarded as microcapsule then subjected to freeze-drying. Please be clear in these aspects and describe step by step, clearly providing rationale for each one of them.

Moisture determination: no need to include the equation.

2.10 Surface morphology by SEM.

This is a qualitative assessment of the microstructure and as such must be acknowledged by authors.

2.15 Statistical analysis

Authors should be aware that statistical significance is p ≤ 0.05 and not p < 0.05. This is a serious flaw and must be amended in the whole manuscript.

Results

In this section, authors must provide a number of discussions on cumulative evidence from other evaluated properties and give place to an overall and sound discussion-conclusion based on the several characteristics evaluated in their manuscript.

Flow properties

These properties should also be discussed in the light of findings on surface morphology since clearly these will influence flowability characteristics.

L369-370 Such differences are difficult to assess by just an empirical observation. If authors want to detect minute differences, they should be prepared to perform a quantitative image analysis on their samples and carefully choose the best morphological descriptor for their capsules and discuss on this basis.

L378 What is a slightly disordered appearance?

Authors captured images by using different magnifications. The comparisons of features made should be based on the same magnification and as such must be declared in materials and methods and consequently systematize in results and discussion.

Conclusions

Once again, authors must provide sound conclusions based on cumulative evidence provided by the various techniques used.

Reviewer 2 Report

This manuscript is interesting and has enough aspects of novelty. Abstract: Please give more specific information about the flowability properties of microcapsules. Please indicate which material is most suitable in the aspect of storage stability.

Line 53-55: Please add a reference to support the point.

Line 137: Please give the meanings of the physicals in Equation 1.

Round 2

Reviewer 1 Report

Authors did a good revision of their manuscripts and responded to my previous concerns. I, therefore recommend this manuscript to be accepted for publication in its present form.